# Synthesis and Characterization of Click Chemical Probes for Single-Cell Resolution Detection of Epichaperomes in Neurodegenerative Disorders

**DOI:** 10.3390/biomedicines12061252

**Published:** 2024-06-04

**Authors:** Sadik Bay, Chander S. Digwal, Ananda M. Rodilla Martín, Sahil Sharma, Aleksandra Stanisavljevic, Anna Rodina, Anoosha Attaran, Tanaya Roychowdhury, Kamya Parikh, Eugene Toth, Palak Panchal, Eric Rosiek, Chiranjeevi Pasala, Ottavio Arancio, Paul E. Fraser, Melissa J. Alldred, Marco A. M. Prado, Stephen D. Ginsberg, Gabriela Chiosis

**Affiliations:** 1Chemical Biology Program, Memorial Sloan Kettering Cancer Center, New York, NY 10065, USA; bays1@mskcc.org (S.B.); digwalc@mskcc.org (C.S.D.); ananda.rodilla@mssm.edu (A.M.R.M.); sharmas5@mskcc.org (S.S.); rodinaa@mskcc.org (A.R.); roychot@mskcc.org (T.R.); parikhk@bxscience.edu (K.P.); eugene.toth@nyu.edu (E.T.); panchap1@mskcc.org (P.P.); pasalac@mskcc.org (C.P.); 2Center for Dementia Research, Nathan Kline Institute, Orangeburg, NY 10962, USA; aleksandra.stanisavljevic@nki.rfmh.org (A.S.); melissa.alldred@nki.rfmh.org (M.J.A.);; 3Department of Physiology and Pharmacology, Schulich School of Medicine, Robarts Research Institute, The University of Western Ontario, London, ON N6A 3K7, Canada; aattara@uwo.ca (A.A.); mprado@robarts.ca (M.A.M.P.); 4Department of Anatomy and Cell Biology, Schulich School of Medicine, Robarts Research Institute, The University of Western Ontario, London, ON N6A 3K7, Canada; 5Molecular Cytology Core, Memorial Sloan Kettering Cancer Center, New York, NY 10065, USA; rosieke@mskcc.org; 6Taub Institute for Research on Alzheimer’s Disease and the Aging Brain, New York, NY 10032, USA; oa1@cumc.columbia.edu; 7Department of Medicine, Columbia University, New York, NY 10032, USA; 8Department of Pathology and Cell Biology, Columbia University, New York, NY 10032, USA; 9Tanz Centre for Research in Neurodegenerative Diseases and Department of Medical Biophysics, University of Toronto, Toronto, ON M5R 0A3, Canada; paul.fraser@utoronto.ca; 10Departments of Psychiatry, NYU Grossman School of Medicine, New York, NY 10016, USA; 11Neuroscience & Physiology & the NYU Neuroscience Institute, NYU Grossman School of Medicine, New York, NY 10016, USA; 12Department of Medicine, Division of Solid Tumors, Memorial Sloan Kettering Cancer Center, New York, NY 10065, USA

**Keywords:** neurodegenerative disease, fluorescence imaging, imaging probe, drug discovery, therapeutic strategy, epichaperomes, click probes

## Abstract

Neurodegenerative disorders, including Alzheimer’s disease (AD) and Parkinson’s disease (PD), represent debilitating conditions with complex, poorly understood pathologies. Epichaperomes, pathologic protein assemblies nucleated on key chaperones, have emerged as critical players in the molecular dysfunction underlying these disorders. In this study, we introduce the synthesis and characterization of clickable epichaperome probes, PU-TCO, positive control, and PU-NTCO, negative control. Through comprehensive in vitro assays and cell-based investigations, we establish the specificity of the PU-TCO probe for epichaperomes. Furthermore, we demonstrate the efficacy of PU-TCO in detecting epichaperomes in brain tissue with a cellular resolution, underscoring its potential as a valuable tool for dissecting single-cell responses in neurodegenerative diseases. This clickable probe is therefore poised to address a critical need in the field, offering unprecedented precision and versatility in studying epichaperomes and opening avenues for novel insights into their role in disease pathology.

## 1. Introduction

Neurodegenerative conditions, including Alzheimer’s disease (AD), Parkinson’s disease (PD), and related central nervous system (CNS) disorders, represent complex, currently untreatable neurodegenerative disorders [1,2]. Various stressors, including age, gender-related changes, proteotoxic insults, and genetic and environmental factors, collectively damage vulnerable cells and brain circuitry over decades [3,4,5]. However, the mechanisms underlying selective vulnerability from cells to networks in these disorders remain poorly understood, hampering both disease understanding and therapeutic development. Consequently, there is an urgent need for innovative tools and technologies for unravelling the molecular and cellular intricacies underlying the early stages of neurodegeneration.

Epichaperomes, intricate long-lived assemblies nucleating on key chaperone proteins such as heat shock protein 90 (HSP90) and heat shock cognate 70 (HSC70) [6], play a pivotal role in the decline in brain functions observed in various CNS disorders, including AD [7,8], PD [9], and traumatic brain injury [10]. By forming scaffolding platforms, epichaperomes sequester and reshape the interaction of proteins crucial for neuronal function, thus contributing to cognitive decline [4,6,8,11]. Notably, the functional reversal of disease-related phenotypes by epichaperome disruptors underscores their critical role in regulating functions underlying disease pathology, suggesting a novel therapeutic approach [7,8,9,12]. In mouse studies, epichaperome formation precedes pathological manifestations, suggesting their potential as early indicators of neurodegeneration [8,13,14,15]. 

Distinguished from chaperones, HSP90 and HSC70 within epichaperomes exhibit a ‘conformational mutation’, as their structure is altered by pathological post-translational modifications (PTMs) [6,16,17]. These PTMs enhance HSP90’s interactions with other chaperones and co-chaperones, creating a microenvironment conducive to the assembly of epichaperomes. Consequently, epichaperomes can be described as ‘assembly mutants’ due to chaperones’ prolonged interactions with other proteins, differing from the dynamic assemblies seen in normal, physiological chaperones. Exploiting these differences, drug candidates targeting epichaperomes have been developed, such as PU-H71 (zelavespib, Phase 1–2 in cancer) [18,19] and PU-AD (icapamespib, Phase 2 in AD) [8,20,21], which target epichaperomes by binding to HSP90, and LSI-137 and YK5 (late preclinical stage), which disrupt epichaperomes by binding to another epichaperome component, HSC70 [22].

The unique role of epichaperomes as both pharmacologically actionable and non-invasively imageable targets in AD and related disorders has driven a critical need for the development of probes and protocols for detecting, quantifying, and studying epichaperomes across diverse biological contexts, such as in cells, tissues, and whole organisms. In our pursuit of understanding epichaperome biology, our team has developed a range of chemical tools and methodologies. These include a radiolabeled [^124^I]-PU-AD probe for epichaperome detection by positron emission tomography (PET) in mice and humans [20,23], a size-based native Western blot method for epichaperome detection in cell or tissue homogenates [8,24], and a chemoproteomics method termed epichaperomics or dysfunctional Protein–Protein Interactome (dfPPI), which identifies proteins and functions impacted by epichaperomes, allowing for the delineation of processes affected by epichaperome formation in disease [8,11]. However, while these tools have significantly contributed to our understanding of epichaperomes in disease biology, they often lack the requisite sensitivity, cellular resolution, and versatility necessary for addressing fundamental questions pertaining to epichaperome dynamics and composition.

For visualizing epichaperomes at the cellular level, a fluorescein isothiocyanate (FITC)-labeled epichaperome probe, PU-FITC, was developed [25]. Designed for the flow cytometry-based detection and quantification of epichaperomes, PU-FITC has been successfully used in analyzing plasma and bone marrow samples [19,26]. However, using FITC with brain biospecimens presents challenges. FITC-labeled probes may nonspecifically bind to various components within brain tissue, resulting in background fluorescence and a reduced signal-to-noise ratio [27], thereby complicating the interpretation of the results. Furthermore, autofluorescence in brain tissue can interfere with the detection of fluorescence signals from FITC-labeled probes, particularly in high-sensitivity imaging applications [28,29]. Additionally, directly conjugated small-molecule probes may have limited cell and tissue permeability and target accessibility due to their bulkiness.

To address these challenges, we here embark on developing a clickable epichaperome probe with a reactive moiety suitable for bioorthogonal chemical ligation with a fluorescent reporter. This design enables delivery to live cells, tissues, or animals, followed by in situ attachment of the fluorescent tag via click chemistry [30]. A diverse range of commercially available fluorophores, including red, far-red, and near-infrared dyes [31], can serve as the fluorescent reporter. Through this design, these probes offer superior tissue penetration and enhanced sensitivity, crucial for detecting epichaperomes within complex biological environments. In this report, we present the synthesis and characterization of the epichaperome click probe, along with a relevant negative control. We demonstrate target selectivity in cells and brain tissue and provide proof-of-principle for their use in detecting and imaging epichaperomes in neurodegenerative disorders, at the single-cell level, in murine brain specimens.

## 2. Methods

### 2.1. Cell Lines and Culture Conditions

The human cell lines MDA-MB-468 (HTB-132, RRID: CVCL_0419) and ASPC1 (CRL-1682, RRID: CVCL_0152) and the colon fibroblasts CCD-18 (CRL-1459, RRID: CVCL_2379) were obtained from American Type Culture Collection (ATCC) (Manassas, VA, USA) and cultured according to the provider’s recommended conditions. Authentication was performed using short tandem repeat profiling, and the cells were regularly tested for mycoplasma contamination.

### 2.2. Mouse Models

M83 transgenic mice carrying the human A53T mutation in the SNCA gene, controlled by the mouse prion promoter (M83 line, B6;C3-Tg(Prnp-SNCA*A53T)83Vle/J, RRID:IMSR_JAX:004479) [32,33], were acquired from The Jackson Laboratory, Farmington, CT, USA. M83 mice are a transgenic mouse model used in research on synucleinopathy, such as PD and Lewy Body dementia, specifically focusing on the pathological mechanisms associated with α-synuclein aggregation. The M83 homozygous mice with two copies of the transgene exhibit several notable deficits starting around 10 months of age [34,35]. Initially, they display hyperactivity, followed by severe motor impairments characterized by wobbling, posturing, decreased locomotor activity, and stiffness of the tail. Ultimately, they progress to an inability to right themselves, leading to end-stage disease within 14–21 days of onset [36]. This phenotype is variable, however, in terms of penetrance. Recent work has shown that homozygous M83 mice have behavioral flexibility deficits with high penetrance that can be detected after 9 months of age and can be corrected by manipulations that decrease epichaperomes [35]. Neuropathologically, the affected mice display S129 phosphorylation of α-synuclein in various brain regions, accompanied by ubiquitin and phosphorylated neurofilament-H accumulation [37,38]. The accumulation of aggregated and insoluble S129 phosphorylated α-synuclein in these mice follows a similar accumulation as that observed in humans with synucleinopathy. Reactive gliosis and mitochondrial alterations are also evident in various brain regions, along with synaptic dysfunction in the hippocampus, which may contribute to cognitive deficits [39,40]. These mice were bred on a C57BL/C3H background to produce transgenic homozygous and wildtype littermates. The mice were maintained in ventilated plexiglass cages and provided with food (Harlan) and water ad libitum. The rooms were maintained at a controlled temperature (22–25 °C) and humidity (40–60%), following a light/dark cycle from 7 a.m. to 7 p.m. All mice in all studies were observed for clinical signs at least once daily.

### 2.3. Reagents 

All commercial chemicals and solvents were of reagent grade and used without further purification. PU-H71, PU-TCO, and PU-NTCO were synthesized following published protocols [22,41] and the procedures described below. Column chromatography, analytical thin layer chromatography (TLC) on 250 μM silica gel F254 plates, preparative TLC on 1000 μM silica gel F254 plates, and flash chromatography using a CombiFlash^®^Rf instrument (Teledyne ISCO, Inc., Chicago, IL, USA) were performed during the synthesis. The identity and purity of each intermediate and the final product were determined by mass spectrometry (MS), High-Performance Liquid Chromatography (HPLC), TLC, and Nuclear Magnetic Resonance (NMR) analysis. Low-resolution mass spectra were acquired using a Waters Acquity Ultra Performance LC equipped with electrospray ionization and an SQ detector (Waters Corporation, Milford, CT, USA). ^1^H/^13^C NMR spectra were obtained on a Bruker 600 MHz instrument (Bruker, Billerica, MA, USA), where chemical shifts were denoted in δ values in ppm relative to TMS as the internal standard. For ^1^H data, details included the chemical shift, multiplicity (s = singlet, d = doublet, t = triplet, q = quartet, br = broad, m = multiplet), coupling constant (Hz), and integration. Similarly, ^13^C chemical shifts were reported in δ values in ppm relative to TMS as the internal standard. The purity of the target compounds exceeded 95%, as determined by Liquid Chromatography/Mass Spectrometry on a Waters Acquity Ultra-Performance Liquid Chromatography (UPLC) system equipped with a Photodiode Array, a MicroMass Single Quadrupole, and Evaporative Light Scattering detectors, using a reversed-phase column (Acquity UPLC BEH C18 column, 2.1 mm × 100 mm, 1.7 μm) eluted with water/acetonitrile gradients containing 0.1% trifluoroacetic acid (TFA). Stock solutions of all chemical probes were prepared in molecular biology grade dimethyl sulfoxide (DMSO) (Sigma Aldrich, St. Louis, MO, USA) at 1000× concentrations.

#### 2.3.1. Synthesis of PU-TCO (4)

PU-TCO (**4**) was synthesized as reported previously [22]. Briefly, the coupling reaction of the free aliphatic amine of **1** and 4-(tertbutoxycarbonylamino)butyric acid with dicyclohexylcarbodiimide (DCC), followed by the removal of the boc-group, provided intermediate **3**. The final product **4** with the TCO click handle was synthesized in a single step by the conjugation of **3** with the commercially available activated ester of TCO. ^1^H NMR (600 MHz, CDCl_3_/CD_3_OD): δ 8.25 (s, 1H), 7.38 (s, 1H), 7.05 (s, 1H), 6.05 (s, 2H), 5.60–5.44 (m, 2H), 4.30 (dd, *J* = 10.2, 5.7 Hz, 1H), 4.25 (t, *J* = 6.8 Hz, 2H), 3.25–3.15 (m, 4H), 2.36–2.25 (m, 5H), 2.03–1.96 (m, 3H), 1.96–1.81 (m, 5H), 1.77–1.67 (m, 2H), 1.58–1.48 (m, 1H); ^13^C NMR (150 MHz, CDCl_3_/CD_3_OD): 173.6, 156.8, 154.3, 152.4, 151.3, 149.9, 149.4, 147.7, 134.9, 132.9, 125.5, 119.6, 119.4, 113.9, 102.6, 94.3, 80.6, 41.1, 41.0, 40.2, 38.6, 35.9, 34.2, 33.7, 32.5, 30.9, 28.9, 25.9; MS (Electrospray Ionization, ESI) *m*/*z* 708.4 [M + H]^+^; HPLC: (a) H_2_O + 0.1% TFA, (b) acetonitrile (ACN) + 0.1% TFA (5 − 95% ACN in 8 min at 0.3 mL/min) Rt = 3.68 min, 99.27%.

#### 2.3.2. Synthesis of 8-((2-Methoxyethyl)thio)-9H-purin-6-amine (6) 

To a suspension of 8-mercaptopurine (**5**; 441 mg, 2.64 mmol) in 1-iodo-2-methoxyethane (573 mg, 3.08 mmol), aqueous KOH (1.5 M, 2.2 mL) was added, and the resulting mixture was stirred at room temperature (rt) for 24 h. The reaction mixture was concentrated under reduced pressure and the resulting residue was purified by column chromatography (CH_2_Cl_2_:CH_3_OH; 100:0 to 85:15) to yield 520 mg (87%) of **6**. ^1^H NMR (600 MHz, DMSO-*d_6_*) δ 13.01 (br s, 1H), 8.05 (s, 1H), 7.01 (br s, 2H), 3.62–3.64 (m, 2H), 3.44–3.46 (m, 2H), 3.28 (s, 3H); ^13^C NMR (125 MHz, DMSO-*d_6_*) δ 154.34, 152.64, 152.00, 147.20, 119.74, 71.00, 58.39, 31.10; MS (ESI) *m*/*z* 225.9 [M + H]^+^.

#### 2.3.3. Synthesis of *tert*-Butyl (3-(6-Amino-8-((2-methoxyethyl)thio)-9H-purin-9-yl)propyl)carbamate (7)

To a stirred solution of **6** (500 mg, 2.22 mmol) in dimethylformamide (DMF) (5 mL), 3-(boc-amino)propyl bromide (793 mg, 3.33 mmol) and Cs_2_CO_3_ (1080 mg, 3.33 mmol) were added, and the mixture was stirred at rt for 6 h. The solvent was removed under reduced pressure, and the resulting residue was purified by preparatory TLC (CH_2_Cl_2_:MeOH:AcOH, 300:2:4) to yield 467 mg (55%) of **7**. ^1^H NMR (600 MHz, CDCl_3_) δ 8.26 (s, 1H), 5.86 (br s, 2H), 5.70 (br s, 1H), 4.19 (t, *J* = 6.0 Hz, 2H), 3.74 (t, *J* = 5.8 Hz, 2H), 3.55 (t, *J* = 5.8 Hz, 2H), 3.40 (s, 3H), 3.03 (s, 2H), 1.96 (m, 2H), 1.46 (s, 9H); ^13^C NMR (125 MHz, CDCl_3_) δ 156.02, 153.44, 152.35, 151.70, 149.57, 119.59, 79.18, 70.72, 58.82, 40.04, 36.79, 31.78, 29.19, 28.44; MS (ESI) *m*/*z* 383.2 [M + H]^+^.

#### 2.3.4. Synthesis of 9-(3-Aminopropyl)-8-((2-methoxyethyl)thio)-9H-purin-6-amine (8)

To a stirred solution of **7** (450 mg, 0.849 mmol) in dry CH_2_Cl_2_ (5 mL), TFA (500 μL) was added, and the solution was stirred at rt for 3 h. The solvent was removed under reduced pressure and the residue was purified by the preparatory TLC (CH_2_Cl_2_:MeOH, 10:1) to yield 282 mg (85%) of **8**. ^1^H NMR (600 MHz, CDCl_3_/CD_3_OD) δ 8.22 (s, 1H), 4.26 (t, *J* = 6.6 Hz, 2H), 3.74 (t, *J* = 5.8 Hz, 2H) [merged in moisture], 3.56 (t, *J* = 5.9 Hz, 2H), 3.41 (s, 3H), 2.95 (t, *J* = 7.2 Hz, 2H), 2.18–2.23 (m, 2H); ^13^C NMR (125 MHz, CDCl_3_/CD_3_OD) δ 152.02, 150.96, 150.58, 146.14, 118.74, 70.23, 58.67, 40.06, 36.33, 31.71, 26.66; MS (ESI) *m*/*z* 283.1 [M + H]^+^.

#### 2.3.5. Synthesis of *tert*-Butyl (4-((3-(6-Amino-8-((2-methoxyethyl)thio)-9H-purin-9-yl)propyl)amino)-4-oxobutyl) carbamate (9) 

A mixture of **8** (250 mg, 0.886 mmol), 4-((*tert*-butoxycarbonyl)amino)butanoic acid (272 mg, 1.335 mmol), DCC (365 mg, 1.768 mmol), and a catalytic amount of 4-(Dimethylamino)pyridine (DMAP) (10.6 mg, 0.089 mmol) in CH_2_CI_2_ (5 mL) was stirred at rt overnight. The reaction mixture was concentrated under reduced pressure and the resulting residue was purified by preparatory TLC (CH_2_Cl_2_:MeOH-NH_3_ (7N), 20:1) to give 249 mg (60%) of **9**. ^1^H NMR (600 MHz, CD_3_OD) δ 8.14 (s, 1H), 4.19 (t, *J* = 7.1 Hz, 2H), 3.75 (t, *J* = 6.0 Hz, 2H), 3.56 (t, *J* = 6.0 Hz, 2H), 3.39 (s, 3H), 3.22 (t, *J* = 6.5 Hz, 2H), 3.09 (t, *J* = 6.7 Hz, 2H), 2.24 (t, *J* = 7.5 Hz, 2H), 2.01 (p, *J* = 6.7 Hz, 2H), 1.79 (p, *J* = 7.0 Hz, 2H), 1.44 (s, 9H); ^13^C NMR (125 MHz, CD_3_OD) δ 174.25, 157.12, 154.07, 151.30, 151.18, 149.54, 119.04, 78.53, 70.47, 57.56, 40.50, 39.47, 36.12, 33.03, 31.35, 28.62, 27.37, 25.85; MS (ESI) *m*/*z* 468.4 [M + H]^+^.

#### 2.3.6. Synthesis of 4-Amino-N-(3-(6-amino-8-((2-methoxyethyl)thio)-9H-purin-9-yl)propyl)butanamide (10)

To a stirred solution of **9** (230 mg, 0.492 mmol) in dry CH_2_Cl_2_ (5 mL), TFA (500 μL) was added. The solution was stirred at rt for 3 h. The solvent was removed under reduced pressure and the residue was purified by preparatory TLC (CH_2_Cl_2_:MeOH, 10:1) to yield 145 mg (80%) of **10**. ^1^H NMR (600 MHz, CD_3_OD) δ ^1^H NMR (500 MHz, MeOD) δ 8.14 (s, 1H), 4.19 (t, *J* = 7.2 Hz, 2H), 3.75 (t, *J* = 6.1 Hz, 2H), 3.57 (t, *J* = 6.0 Hz, 2H), 3.39 (s, 3H), 3.23 (t, *J* = 6.7 Hz, 2H), 2.92 (t, *J* = 7.4 Hz, 2H), 2.36 (t, *J* = 7.1 Hz, 2H), 2.02 (p, *J* = 6.9 Hz, 2H), 1.90 (p, *J* = 7.2 Hz, 2H); ^13^C NMR (125 MHz, CD_3_OD) δ 173.45, 154.08, 151.28, 151.15, 149.54, 119.04, 70.45, 57.56, 40.49, 39.41, 36.23, 32.52, 31.39, 28.59, 24.30; MS (ESI) *m*/*z* 368.5 [M + H]^+^.

#### 2.3.7. Synthesis of (E)-Cyclooct-4-en-1-yl (4-((3-(6-amino-8-((2-methoxyethyl)thio)-9H-purin-9-yl)propyl) amino)-4-oxobutyl)carbamate (PU-NTCO, 11)

A mixture of **10** (15 mg, 0.029 mmol), TCO-NHS ester (9.3 mg, 0.035 mmol), and triethylamine (9 μL, 0.058 mmol) in DMF (1 mL) was stirred at rt for 3 h in the dark. The reaction mixture was concentrated under reduced pressure and the resulting residue was purified by flash chromatography (CH_2_Cl_2_:MeOH, 20:1) to give 13.7 mg (65%) of compound **11** (PU-NTCO). ^1^H NMR (600 MHz, CDCl_3_/CD_3_OD) δ 8.22 (s, 1H), 5.60–5.44 (m, 2H), 4.34–4.27 (m, 1H), 4.15 (t, *J* = 6.8 Hz, 2H), 3.75 (t, *J* = 6.0 Hz, 2H), 3.53 (t, *J* = 6.0 Hz, 2H), 3.41 (s, 3H), 3.22–3.11 (m, 4H), 2.38–2.26 (m, 5H), 2.03–1.80 (m, 8H), 1.77–1.65 (m, 2H), 1.57–1.48 (m, 1H); ^13^C NMR (150 MHz, CDCl_3_/CD_3_OD) δ 173.41, 156.72, 153.53, 151.83, 151.67, 149.94, 134.97, 133.00, 119.28, 80.60, 70.55, 58.91, 53.49, 41.19, 40.35, 38.67, 35.77, 34.31, 33.77, 32.54, 31.81, 30.97, 28.47, 25.90, 0.00.; MS (ESI) *m*/*z* 520.3 [M + H]^+^; HPLC: (a) H_2_O + 0.1% TFA, (b) ACN + 0.1% TFA (5 − 95% ACN in 8 min at 0.3 mL/min) R_t_ = 3.14 min, 97.52%.

### 2.4. Fluorescence Polarization Assay

For the binding studies, fluorescence polarization (FP) assays were conducted following previously reported methods [42]. MDA-MB-468 cancer cell lysates were prepared in a native buffer containing 20 mM Tris pH 7.4, 20 mM KCl, 5 mM MgCl_2_, and 0.01% NP40, and the protein concentration was determined using the BCA assay according to the manufacturer’s instructions (Pierce™ BCA Protein Assay Kit, Thermo Fisher Scientific, Waltham, MA, USA). Stock solutions of 10 μM PU-FITC, 1 mM PU-H71, 1 mM PU-TCO, and 1 mM PU-NTCO were prepared in DMSO and diluted with Felts buffer (20 mM Hepes (K), pH 7.3, 50 mM KCl, 2 mM dithiothreitol (DTT), 5 mM MgCl_2_, 20 mM Na_2_MoO_4_, and 0.01% NP40 with 0.1 mg/mL Bovine Gamma Globulin (BGG). PU-TCO, PU-NTCO, and PU-H71 were added at various concentrations to the assay buffer containing both 6 nM PU-FITC and the cell lysate (12.5 µg/well) in a final volume of 100 µL in black 96-well microplates (Corning, #3650, Corning, NY, USA). Test compounds were applied to triplicate wells, with each assay plate comprising background wells (buffer only), tracer controls (free PU-FITC only), and bound controls (PU-FITC in the presence of protein). Following a 24 h incubation period on a shaker at 4 °C, FP values (in mP) were determined using an Analyst GT instrument (Molecular Devices, Sunnyvale, CA, USA). The fraction of PU-FITC bound to epichaperomes was correlated with the mP value and plotted against competitor concentrations. For FITC, excitation was filtered at 485 nm and emission at 530 nm, utilizing a dichroic mirror of 505 nm. The percentage binding was computed using the following formula: (% Control) = ((mPc − mPf)/(mPb − mPf)) × 100, where mPc represents the recorded mP from compound wells, mPf is the average recorded mP from PU-FITC–only wells, and mPb denotes the average recorded mP from wells containing both PU-FITC and lysate. All experimental data underwent analysis using SOFTmax Pro 4.3.1, and the inhibitor concentration at which 50% of the bound PU-FITC was displaced was determined by fitting the data via nonlinear regression analysis using Prism 10.0 (GraphPad Software, Boston, MA, USA).

### 2.5. Native or SDS PAGE Coupled with Immunoblotting

MDA-MB-468 and ASPC1 cells were plated in 10 cm plates at 4–6 × 10^6^ cells per plate and treated the next day with 1 or 2 µM PU-TCO for 1 h. The cells were scraped, and the cell suspension was centrifuged at 1200 rpm for 5 min at 4 °C. The cells were lysed in lysis buffer containing 20 mM Tris pH 7.4, 20 mM KCl, 5 mM MgCl_2_, and 0.01% NP40 using a freeze–thaw procedure (repeated three times). The resulting homogenates were centrifuged at maximum speed for 10 min at 4 °C, and the supernatant was carefully transferred to a pre-cooled tube. The protein concentrations in the lysates were determined using the BCA kit (Pierce, #23225) according to the manufacturer’s instructions. Protein was loaded onto a 4–10% native gradient gel, resolved at 4 °C for 2–3 h, and then transferred onto nitrocellulose membranes in Tris-Glycine-Methanol transfer buffer at 4 °C overnight. The membranes were immunoblotted using primary antibodies against HSP90α (Abcam, Cambridge, UK; ab2928, RRID:AB_303423, 1:6000), HOP (Enzo Biochem, Farmingdale, NY, USA; SRA-1500, RRID:AB_10618972, 1:2000), and CDC37 (Cell Signaling, Danvers, MA, USA; 4793, RRID:AB_10695539, 1:3000), followed by incubation with appropriate horseradish peroxidase (HRP)-conjugated secondary antibodies purchased from SouthernBiotech, Birmingham, AL, USA—goat anti-rabbit Ig, human ads-HRP (4010-05, Lot# A4211-ZH10E, 1:5000), and goat anti-mouse IgG, human ads-HRP (1030-05, Lot# D1922-X922, 1:5000)). The same lysates were analyzed by Western blotting. The cell lysates were denatured in the presence of sodium dodecyl sulfate (SDS) at 85 °C. A total of 10 to 15 μg of the total protein was resolved on SDS-PAGE and then transferred to nitrocellulose membranes, blocked for 1 h in 5% milk in Tris-buffered saline (TBS), and incubated overnight with the primary antibodies mentioned above, followed by a horseradish peroxidase (HRP)-conjugated secondary antibody (see above). β-actin (A1978, Sigma-Aldrich, RRID: AB_476692, 1:3000) was used to control for protein loading.

### 2.6. Epichaperome Imaging in Cells in the Culture 

For microscopy, the cells were seeded onto four-well glass plates (Cellvis, #C4-1.5H-N; Mountain View, CA, USA) and cultured in Dulbecco’s Modified Eagle Medium (DMEM) supplemented with 5% FBS (Thermo Fisher Scientific, A3840001) at 37 °C in a 5% CO_2_ environment. MDA-MB-468 cells were seeded at a density of 2.5 × 10^4^ per well, ASPC1 cells were seeded at 10 × 10^4^ per well, and CCD-18 cells were seeded at 5 × 10^4^ per well 48 h prior to the experiment to achieve 60–70% confluency on the day of the experiment. To detect epichaperomes, cells were treated with 1 µM PU-TCO or 1 µM PU-NTCO for 1 h at 37 °C. Following a rinse in DMEM and Phosphate Buffered Saline (PBS), the samples were fixed with 4% paraformaldehyde (PFA, Thermo Fisher Scientific, 043368.9M) at rt for 15 min and then permeabilized using 0.2% Triton X-100 (MP Biomedicals, #194854; Santa Ana, CA, USA) in PBS for 15 min at rt. A click reaction was performed using 700 nM Cy5 Tetrazine (Click Chemistry Tools, #1189-1, Scottsdale, AZ, USA) in PBS for 12 min at rt. The cells were rinsed twice with PBS following the click reaction. Subsequently, the cells were blocked with 1% Bovine Serum Albumin (BSA)/PBS for 30 min at rt and then incubated for 1 h at rt with an HSP90α (Abcam, ab2928, RRID:AB_303423, 1:500) primary antibody (prepared in blocking buffer). After three washes using PBS, the Alexa Fluor 488 conjugated secondary antibody (Thermo Fisher Scientific, A-11008, RRID:AB_143165, 1:500) was added to the chambers for a 3 h incubation at rt. The cells were then rinsed three times with PBS. Finally, the cells were stained with 200 nM Hoechst 33342 (Thermo Fisher Scientific, 62249) in PBS for 5 min to visualize the nuclei. Images of the stained cells were captured using the Zeiss LSM880 microscope with Zeiss plan-apochromat 20×/NA 0.8 objectives (Zeiss Inc., Oberkochen, Germany).

### 2.7. Epichaperome Detection in Brain Tissue Sections

For the detection of epichaperomes in the brain, mice were administered a lethal overdose of ketamine (100 mg/kg) and xylazine (20 mg/kg) in 0.9% sterile saline for anesthesia. The unconscious and non-responsive animals were transcardially perfused with ice-cold 1× PBS [35]. Following the brain removal, they were quickly frozen on dry ice and stored at −80 °C until their use. Both male and female mice were used for method development and specificity validation. The frozen brains were sectioned sagittally into 20 μm slices. All probes, in SuperBlock Blocking Buffer (Thermo Fisher Scientific #37515), were added to the brain slices for 1 h at rt as follows: PU-NTCO (control chemical inert probe) at 1 μM and PU-TCO (epichaperome probe) at three different concentrations (0.1, 0.5, and 1 μM). For competitive binding, the slices were first treated for 1 h with 1 μM PU-H71 prior to the PU-TCO addition. The tissue sections were then washed twice with PBS for 15–30 s and then fixed with 4% PFA in PBS for 30 min at rt. After fixation, the slices were washed twice with PBS for 15–30 s and then permeabilized using 0.02% Triton-X100 in PBS for 15 min. Subsequently, the slices were washed twice with PBS for 15–30 s and incubated with the fluorescent reporter (Cy5 Tetrazine, 700 nM diluted in SuperBlock Blocking Buffer) for 15 min at rt. After another round of washing with PBS, the slices were blocked using SuperBlock Blocking Buffer containing 0.3% Triton-X100 for 1 h at rt. Following blocking, the sections were incubated overnight at rt with the primary NeuN antibody (Anti-NeuN [1B7], Abcam, ab104224, RRID: AB_10711040) diluted at 1:500 in SuperBlock Blocking Buffer containing 0.1% Triton-X100. After washing twice with PBS for 15–30 s, the slices were incubated for 3 h at rt with the secondary antibody Alexa Fluor 488 Plus (Thermo Fisher Scientific, #A32723) diluted 1:400 with SuperBlock Blocking Buffer containing 0.1% Triton-X100. Nuclear staining was performed for 15 min at rt using Hoechst (ThermoFischer Scientific, #62249, 1:1000 dilution in PBS). Following PBS washing, the sections were mounted with Glycerol/PBS. The slides were then imaged on a Pannoramic Scanner (3DHistech, Budapest, Hungary) under a 20×/0.8NA objective. High-resolution images of the tissue sections were taken on a Zeiss LSM880 microscope with AiryScan using the oil-immersion Zeiss plan-apochromat 63×/NA 1.4 objective.

## 3. Results

### 3.1. Synthesis and Characterization of Epichaperome Click Probes 

For the synthesis and characterization of the epichaperome click probes, we utilized PU-H71 (zelavespib) and PU-AD (icapamespib) as our starting points (Figure 1a). These small molecules have demonstrated specificity and activity against epichaperomes in various cellular, tissue, and organismal contexts, in both murine and human models [6,20,43]. They exhibit a unique property of becoming kinetically trapped within epichaperome-bound HSP90 while rapidly dissociating from physiologic HSP90 forms found in normal tissues [44]. This property enables discrimination between pathologic and physiologic HSP90 conformers, making them ideal candidates for designing clickable epichaperome probes for imaging.

In our approach, we focused on functionalizing the N9 position of these small molecules based on our prior experience and existing literature [45,46]. We anticipated that incorporating an approximately 8–10 atom spacer between N9 and the click handle would retain target affinity and specificity. While numerous chemistries exist for chemical ligation, the inverse electron-demand Diels–Alder (iEDDA) cycloaddition click reaction between trans–cyclooctene (TCO) and tetrazine [47] was selected for our purposes. This reaction offers high selectivity and fast kinetics, proceeds under mild conditions, and is catalyst-free, making it ideal for both in vitro and in vivo applications [48]. With these considerations in mind, we focused on synthesizing a probe that incorporates the PU-AD/PU-H71 pharmacophore, featuring a linker attached at N9 and containing a TCO moiety for in situ probe labeling with a fluorescence dye (PU-TCO, Figure 1a). Additionally, we designed a negative control probe containing a chemically similar but epichaperome-inactive moiety (PU-NTCO, Figure 1a).

The process of illuminating epichaperomes with the PU-TCO probe involves several steps. First, upon entering the cell, the probe should bind preferentially to a core epichaperome component, HSP90, avoiding binding to the abundant HSP90 pools involved in physiological chaperone folding functions. Once bound to its target, the epichaperome probe undergoes an in situ reaction with a fluorescent dye containing a tetrazine functionality, known as the click step. This reaction converts the probe into a fluorescent entity, which can then be detected using various methods (Figure 1b).

We synthesized the epichaperome probe following the synthetic route described in Figure 2. Key intermediates were synthesized as previously described [49,50,51]. Initially, 3-(boc-amino)-1-propanol was dehydratively coupled with 8-((6-iodobenzo[d][1,3]dioxol-5-yl)thio)-9H-purin-6-amine in the presence of PPh_3_ and DEAD, yielding a boc-protected intermediate. Boc-deprotection with TFA followed, and the resulting free aliphatic amine was then coupled with 4-(tertbutoxycarbonylamino)butyric acid. The subsequent removal of the boc-group provided an intermediate, which was conjugated with the commercially available activated ester of TCO, as depicted in Figure 2, to yield PU-TCO. For the synthesis of the control probe (PU-NTCO), a similar route was followed, with a modified pharmacophore incorporating a 2-(methoxyethyl)thio group instead of (6-iodobenzo[d][1,3]dioxol-5-yl)thio) at the C-8 position of the adenine ring, as shown in Figure 2. The identity of each compound was confirmed by MS and NMR, and the final products were determined to be ≥ 98% pure, as assessed by HPLC (Supplementary Appendix A).

### 3.2. Specificity of the Probes for Epichaperomes

To evaluate the specificity of the probes for epichaperomes, we first considered that while PU-H71 and PU-AD are highly cell- and tissue-permeable molecules, the addition of a click handle might impact their binding to epichaperomes and the cell uptake. Therefore, our initial step was to assess the probe’s retention of target binding and specificity in vitro, in cell homogenates, and in cellulo, in live cells, using our established assay pipeline (Figure 3a,b) [8,20,22].

We utilized a fluorescence polarization (FP) assay to measure the equilibrium competitive binding of the probe to epichaperomes in cell homogenates obtained from an epichaperome-positive cell line (MDA-MB-468 breast cancer cells) (Figure 3a) [8,20]. The assay involved mixing protein extracts with the PU-FITC epichaperome probe and compounds to be tested, followed by reading the FP signal at equilibrium. Both PU-TCO and the negative control probe PU-NTCO were evaluated, with relevant positive and negative controls (PU-FITC only or homogenate only) included in each assay. PU-TCO exhibited a half-maximal effective concentration—EC_50_—value close to that of PU-H71, suggesting that the linker did not significantly alter the probe’s target binding. Conversely, PU-NTCO showed no measurable activity even at the highest tested concentration of 10 µM, supporting its use as a negative control.

When PU-H71 and PU-AD enter cells, they become entrapped within the epichaperome assemblies, prompting the disassembly of epichaperomes into individual components [8,44]. Consequently, when MDA-MB-468 cells (high in epichaperome) were treated with PU-TCO for 1 h, stabilized epichaperome assemblies were observed on native gels during immunoblotting with antibodies against epichaperome component proteins such as HSP90, CDC37, and HSP-organizing protein (HOP) [8,41]. In contrast, PU-TCO showed a minimal effect in ASPC1 cells, which have chaperone levels akin to MDA-MB-468 cells but have minimal epichaperome levels [22] (Figure 3b). This underscores the productive engagement of PU-TCO with the epichaperome in cells, affirming its on-target activity.

To test the probes’ capability to specifically detect epichaperomes in cells, we utilized confocal microscopy. Both epichaperome-high (MDA-MB-468) and epichaperome-low/negative (ASPC1) cells, with comparable levels of HSP90 and other chaperones [22], were employed. Cells cultured in chamber slides were exposed to the PU-TCO clickable probe for 1 h. Subsequently, the cells underwent fixation, permeabilization, and a click reaction with a cy5 derivative as the fluorescent reporter. Control experiments and blocking assays were conducted to ensure the probes’ specificity, accurately discriminating epichaperomes from other cellular components. For instance, PU-NTCO served as a negative control, featuring a clickable derivative that hinders epichaperome binding. Additionally, a blocking experiment involved pre-treating cells with PU-H71 before incubation with the clickable probe, further confirming the probe’s specificity for epichaperome binding (Figure 4a–c). 

We observed that the fluorescence signal corresponded with the presence of the target—notably, the signal intensity in MDA-MB-468 cells was significantly higher than in ASPC1 cells (Figure 4b,c). The signal was competed off with PU-H71, and it was absent when the control probe was used instead of PU-TCO (Figure 4b,c). 

We next performed immunofluorescence with an HSP90 antibody. The HSP90 antibody detects all cellular pools of HSP90, whether involved in folding functions or incorporated into epichaperome platforms. Conversely, the PU-TCO probe should detect HSP90 only when part of epichaperomes, independent of the cellular concentration of HSP90 (Figure 5a). As expected, staining with an HSP90 antibody, which detects all cellular HSP90 pools, revealed an equal staining intensity and a similar HSP90 concentration in both MDA-MB-468 and ASPC1 cells (Figure 5b,c, HSP90 antibody, green). In contrast, the intensity of PU-TCO staining reflected the presence of HSP90 in epichaperomes and not the total HSP90 pools (Figure 5b,c, PU-TCO epichaperome probe, red). Lastly, CCD-18Co, a non-transformed human fibroblast cell line lacking epichaperomes [22], exhibited the HSP90 signal with the HSP90 antibody but no signal upon PU-TCO probing (Figure 5b,c). 

These diverse lines of evidence collectively support on-target binding and the specificity of PU-TCO for epichaperomes, both in vitro and in cells.

### 3.3. Epichaperome Detection in Murine Brain Tissue

To provide proof-of-principle on the use of PU-TCO to detect cells impacted by epichaperome formation, the M83 homozygous alpha-synuclein mice (expressing the human A53T mutation) were selected [52], as their neurodegenerative phenotype is consistent with epichaperome formation. In these mice, a genetic decrease in the epichaperome component HOP rescues disease-related phenotypes, suggestive of epichaperome-driven toxicity [35]. 

For this study, we selected M83 homozygous mice (Figure 6a) as well as wild-type (WT) mice (Figure 6b) at 13 months of age. The mice were perfused with PBS before brain harvesting, and their brains were frozen for sectioning. This method is particularly useful for preserving the tissue’s native structure and, importantly, the integrity of the epichaperome assemblies, which is crucial for immunofluorescence studies. By perfusing the animal with PBS before sectioning, blood is removed from the vasculature, reducing background fluorescence and improving the clarity of the immunofluorescent signals. Additionally, freezing the tissue allows for precise and consistent sectioning, enabling thin slices for microscopic analysis [53,54]. 

To assess the probe’s ability to specifically detect epichaperomes in brain tissue, we generated several 20 μm slices, which were then incubated with PU-TCO (0.1, 0.5, and 1 µM). After fixation (4% PFA) and permeabilization (0.02% Triton X-100), the cy5 fluorescent reporter was attached via click chemistry. Negative controls included PU-NTCO and blocking by pre-treating slices with PU-H71 before incubation with the clickable probe. In addition, the slices were counterstained with Hoechst to visualize individual brain cells. Antibody staining against NeuN (Neuronal Nuclei), a well-established marker for mature neurons in the CNS, was also performed to distinguish neurons from glia [55].

Through confocal microscopy, we observed that the PU-cy5 fluorescence signal was proportional to the concentration of the added probe. Specifically, the signal intensity in tissues stained with 1 µM PU-TCO was notably higher than that with 0.5 µM PU-TCO, and this signal was higher than that observed with 0.1 µM PU-TCO. The signal was competed off with PU-H71, and it was absent when the control probe PU-NTCO was used instead of PU-TCO. Similarly, no signal was detected in the WT mice, consistent with the reported absence of epichaperomes in normal brain tissue [8,20,22]. These observations collectively affirm the specificity of the signal detected in the brain, indicating its association with epichaperomes.

### 3.4. Single-Cell-Level Epichaperome Illumination in the Brain

We then employed high-resolution microscopy to explore the epichaperome probe’s capability to detect and characterize the presence and intensity of epichaperomes in specific brain cells. The ventral striatum, identifiable on the analyzed slices by the presence of its components (the olfactory tubercle (OT) and nucleus accumbens, along with the islands of Calleja), was chosen for the analysis (Figure 7a) [56]. In rodents, besides the OT, no other structure contains the islands of Calleja [57].

The striatum is mainly inhabited by medium spiny neurons, also known as spiny projection neurons, characterized by their large dendrites predominantly covered with dendritic spines [58]. A recent single-nucleus RNA-seq study conducted specifically on the striatum of M83 mice also identified medium spiny neurons as the most prevalent neuronal population, alongside cholinergic interneurons, immature neurons, and neural progenitor cells [59]. Moreover, the M83 striatum harbors various glial cell types, including oligodendrocytes, oligodendrocyte precursor cells, astrocytes, microglia, endothelial cells, and pericytes [59], underscoring its cellular heterogeneity.

Macroscopic and microscopic examination of the region revealed selective vulnerability to epichaperome formation, with certain subregions and cells exhibiting more pronounced staining than others (Figure 7b). For instance, the staining intensity for epichaperomes in the molecular layer of the OT, also known as Layer 1, was generally higher compared to that of Layer 2 (the “dense” cell layer) and Layer 3 (the “multiform” cell layer) [56]. Intriguingly, some of the highly intense cells lacked NeuN staining, indicating their identity as glial cells. The presence of both neurons and glial cells, positive or negative for epichaperomes, was evident when we zoomed in on specific regions (Figure 7c). While we cannot ascertain the identity of the glial cells without staining for cell-specific markers, their morphology suggests the likelihood of being astrocytes and microglia [60]. Further studies with a cell-type-specific marker will be needed to rigorously identify the identity of the cells. Nonetheless, this study marks the first documented instance of brain cells, other than neurons, exhibiting vulnerability to epichaperome formation.

In summary, the successful detection of the cell-specific vulnerability to epichaperome formation using the click probe highlights its potential as a valuable tool for dissecting the intricate cellular responses underlying neurodegenerative diseases.

## 4. Discussion

The development of the clickable epichaperome probe represents a significant advancement in our ability to study the molecular mechanisms underlying neurodegenerative diseases, particularly those involving epichaperome formation. These probes are poised to increase our ability to not only detect and quantify epichaperomes but also gain insights into their dynamic behavior, composition, and influence on intricate cellular processes. The need for such innovation is rooted in the limitations of current research tools and methodologies. While radiolabeled probes and chemoproteomics have provided valuable insights into epichaperomes [8,18,22,41,43,44], they often lack the precision, cellular resolution, and versatility required to address fundamental questions about these biomolecular structures. Clickable probes, however, offer a groundbreaking solution to this challenge. They will enable precise in vitro and in vivo imaging of epichaperomes in specific brain regions, capitalizing on the specificity of fluorescent probes and their compatibility with immunostaining techniques.

One of the key findings of this study is the successful demonstration of PU-TCO’s specificity for epichaperomes both in vitro and in cells, in both human and murine samples. Through fluorescence polarization assays and confocal microscopy, we showed that PU-TCO selectively binds to epichaperomes in cell homogenates, cultured cells, and brain tissue, with minimal off-target effects. This specificity is crucial for accurately identifying and characterizing epichaperome formation in biological samples, laying the foundation for further investigations into the role of epichaperomes in disease pathology.

Furthermore, our results provide compelling evidence for the feasibility of using PU-TCO to detect epichaperomes in brain tissue with cellular resolution. By employing high-resolution microscopy techniques, we were able to visualize and characterize epichaperome formation in specific brain regions, including the ventral striatum, as well as in specific brain cells, encompassing both neurons and glia. This marks a significant advancement in our ability to study epichaperome biology in complex biological environments and holds promise for future studies investigating the role of epichaperomes in neurodegenerative disorders. Importantly, the versatility of this click probe allows for multiplexing with antibodies against specific disease markers, offering the opportunity to investigate intricate mechanistic details.

Importantly, our study also sheds light on the cellular heterogeneity of epichaperome formation in the brain. We observed differential staining patterns in various brain cell populations, with some cells exhibiting higher epichaperome formation and accumulation than others. Notably, we identified glial cells, in addition to neurons, exhibiting vulnerability to epichaperome formation, marking the first documented instance of such observations. This highlights the complexity of epichaperome biology in the brain and underscores the importance of further research elucidating the role of different cell types in epichaperome-mediated neurodegeneration.

## 5. Conclusions

The development and validation of the PU-TCO clickable epichaperome probe represent a significant step forward in our understanding of epichaperome biology and its implications for neurodegenerative diseases. The specificity, sensitivity, and versatility of the probe make it a valuable tool for studying epichaperomes in various contexts, encompassing both cellular and tissue levels, across murine and human systems. Further studies utilizing this probe in preclinical and clinical settings are warranted to fully realize its potential for diagnosing and treating neurodegenerative disorders.

## Figures and Tables

**Figure 1 biomedicines-12-01252-f001:**
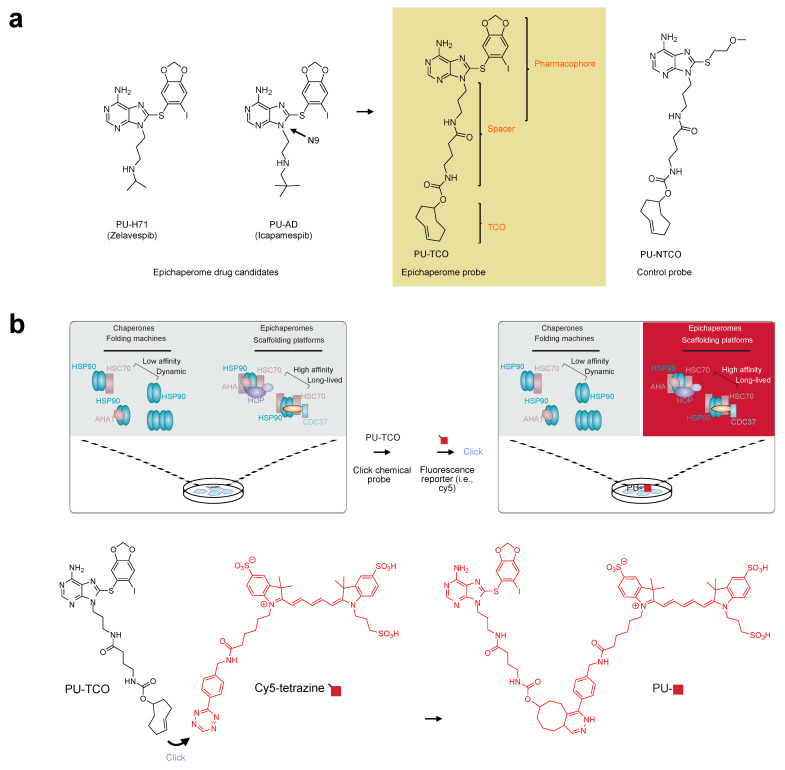
The epichaperome click probe design and principles of epichaperome detection in biological samples. (**a**) Chemical structures of the epichaperome drug candidates (left) and the designed probe, PU-TCO (right). PU-TCO incorporates the pharmacophore of the drug candidates and features a linker attached at N9, along with a trans–cyclooctene (TCO) moiety for in situ probe labeling. A negative control probe, PU-NTCO, containing a chemically similar but epichaperome-inactive moiety, is also depicted. (**b**) The schematic illustrates the concept of epichaperome illumination in biological samples. The probe should effectively permeate into cells and, once inside, interact specifically with epichaperomes by binding to a core component, HSP90, without interfering with the abundant HSP90 pools involved in physiologic chaperone folding functions. Upon in situ reaction with a fluorescent dye carrying a tetrazine functionality—the click step—the probe becomes fluorescent and can be detected by various methods. The schematic demonstrates the specific use of cy5-tetrazine as the fluorescent reporter.

**Figure 2 biomedicines-12-01252-f002:**
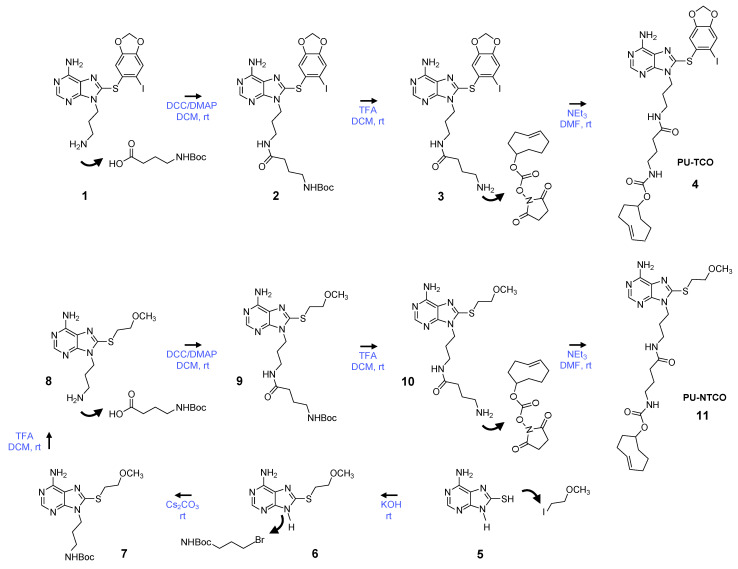
Synthetic scheme for the synthesis of the PU-TCO epichaperome probe and the PU-NTCO control probe. Compound numbering, see Methods, Section 2.3.1, Section 2.3.2, Section 2.3.3, Section 2.3.4, Section 2.3.5, Section 2.3.6 and Section 2.3.7. Abbreviations: DCC, dicyclohexylcarbodiimide; DCM, dichloromethane; DMAP, 4-dimethylaminopyridine; DMF, dimethylformamide; rt, room temperature; NEt3, triethylamine; TFA, trifluoroacetic acid.

**Figure 3 biomedicines-12-01252-f003:**
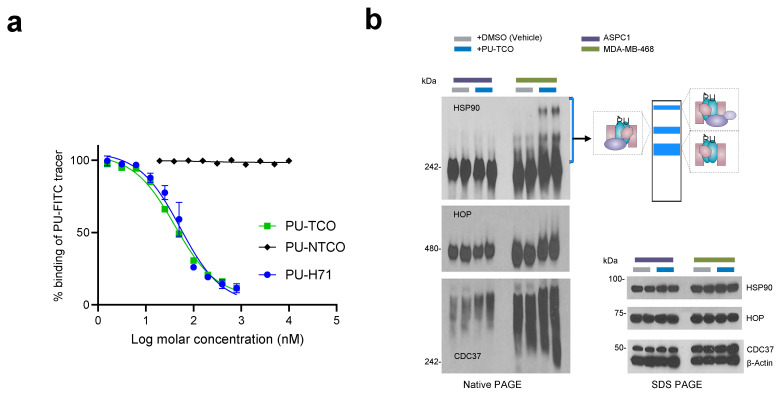
Evaluation of effective epichaperome binding by the designed probes in vitro, in cell homogenates, and in cellulo, in live cells. (**a**) A fluorescence polarization (FP) assay was used to measure the equilibrium competitive binding of the click probes to epichaperomes in cell homogenates obtained from an epichaperome-positive cell line (MDA-MB-468 breast cancer cells). Data are presented as the mean ± s.e.m., n = 3. PU-H71, positive control. (**b**) Detection of epichaperome components (chaperones and co-chaperones HSP90, HOP, and CDC37) through native-PAGE (**left**), followed by immunoblotting in cells treated in duplicate with DMSO (vehicle control) or PU-TCO (1 and 2 µM) for 1 h. Blue brackets indicate the approximate position of epichaperome-incorporated chaperones. Western blotting analysis (**right**, SDS-PAGE) was used to evaluate the total levels of these proteins. β-Actin serves as the protein loading control. Gel images are representative of two independent experiments, with each condition performed at two concentrations of the agent.

**Figure 4 biomedicines-12-01252-f004:**
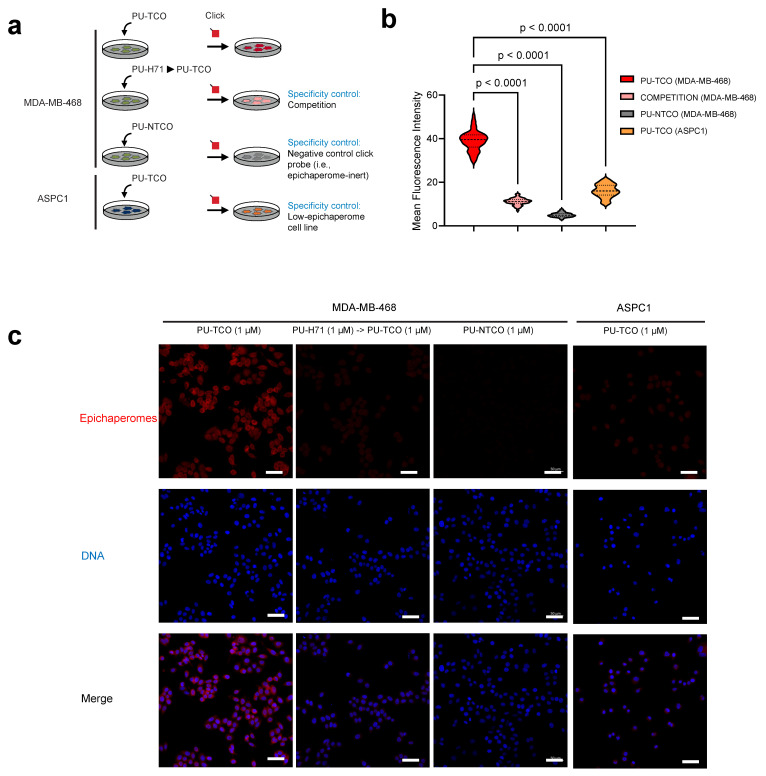
Evaluation of effective and selective epichaperome binding by PU-TCO in cellulo, in live cells. (**a**) Overview of the experimental design showing several control experiments designed to test probe specificity. Epichaperome-high (MDA-MB-468) and epichaperome-low (ASPC1) cancer cell lines were stained with the cy5 fluorescence reporter following the addition of PU-TCO (1 µM), PU-NTCO (1 µM, control epichaperome-inert probe), or PU-TCO after the pretreatment of the cells for 1 h with PU-H71 (1 µM) (competition). (**b**) Graph depicting the median, dotted line, and quartiles, dashed lines; n = 50 cells from 3 replicate experiments; one-way ANOVA with Sidak’s post-hoc. Each data point represents the mean fluorescence intensity recorded per cell. (**c**) Representative micrographs from three individual experiments, per panel (**b**). Images were captured using an LSM880 confocal microscope (Zeiss) with the 20×/0.8NA objective. Scale bars represent 50 µm.

**Figure 5 biomedicines-12-01252-f005:**
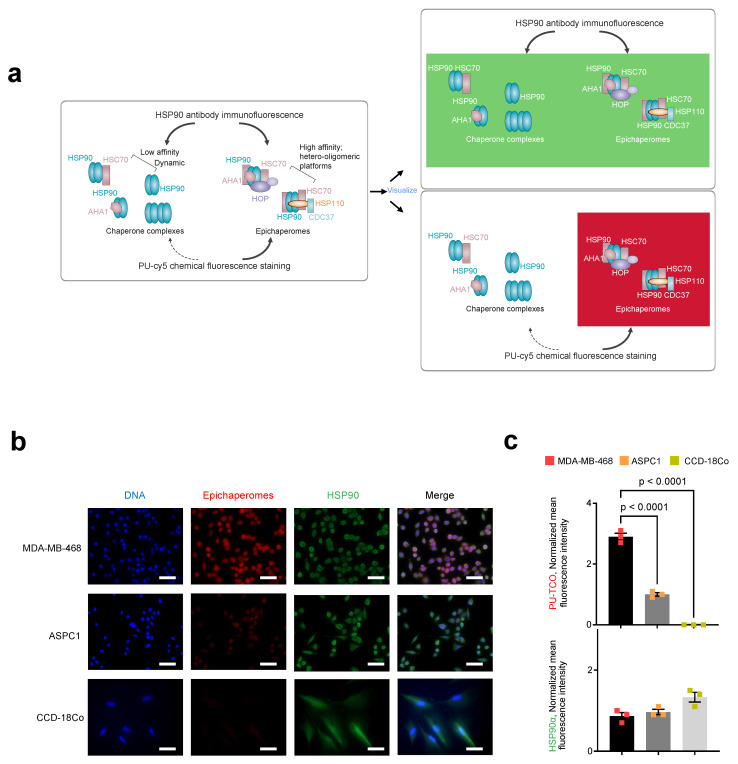
Side-by-side evaluation of selective epichaperome binding by PU-TCO in cellulo compared to immunoblotting with an HSP90 antibody. (**a**) Schematic illustrating the biochemical and functional distinction between chaperones and epichaperomes. The HSP90 antibody detects all cellular pools of HSP90, whether involved in folding functions or incorporated into epichaperome platforms. Conversely, the PU-TCO probe should detect HSP90 only when part of epichaperomes, independent of the cellular concentration of HSP90. Epichaperome-high (MDA-MB-468) cancer cells, epichaperome-low (ASPC1) cancer cell lines and CCD-18Co, non-transformed colon fibroblasts, in culture, were stained with the cy5 fluorescence reporter following the addition of PU-TCO (1 µM), and with an Alexa488-labeled HSP90 antibody. (**b**) Representative micrographs from three individual experiments, per panel (**a**), are shown. Images were captured using an LSM880 confocal microscope (Zeiss) with the 20×/0.8NA objective. Scale bars represent 50 µm. (**c**) The data per panel (**a**) are presented as the mean ± s.e.m., n = 3, one-way ANOVA with Dunnetts’s post-hoc. Each data point represents the average of the mean fluorescence intensity recorded per cell from each experiment.

**Figure 6 biomedicines-12-01252-f006:**
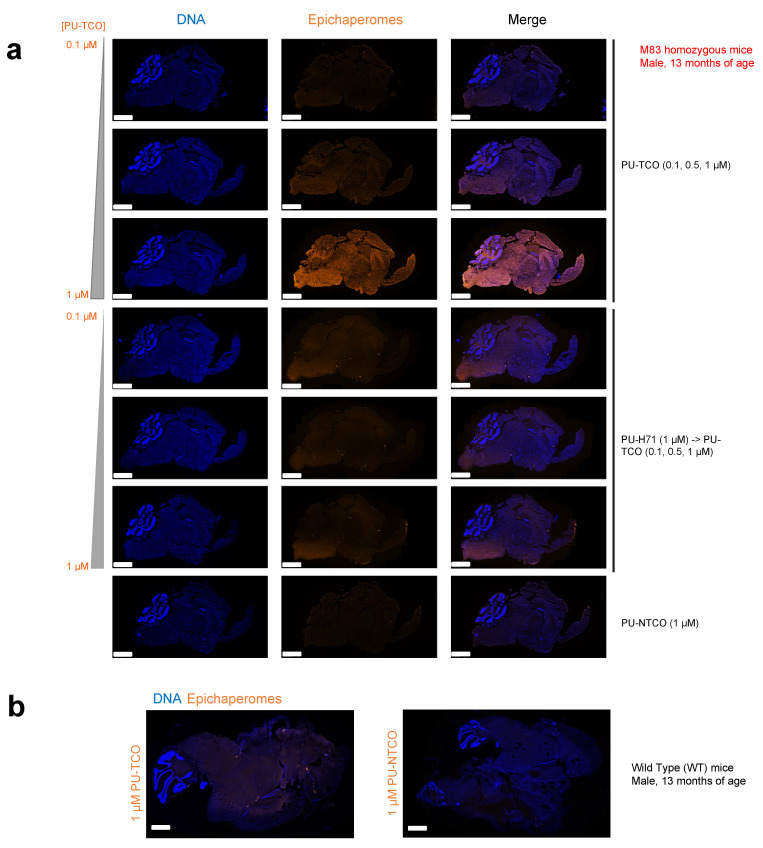
Evaluation of selective epichaperome detection by PU-TCO in post-mortem murine brains. (**a**,**b**) Frozen brains harvested from male M83 homozygous mice (**a**) and wild-type mice (**b**) at 13 months of age were sectioned (20 μm) for staining. Sagittal slices were incubated with PU-TCO (0.1, 0.5, and 1 µM), and then the cy5 fluorescent reporter was attached via click chemistry. Negative controls included PU-NTCO and blocking by pre-treating slices with PU-H71 (1 µM, 1 h) before incubation with the PU-TCO clickable probe. Epichaperomes, orange; Hoechst (blue), for visualization, and staining of cell nuclei. The slides were scanned on a Pannoramic Scanner (3DHistech) using a 20×/0.8NA objective. Scale bars represent 2 mm.

**Figure 7 biomedicines-12-01252-f007:**
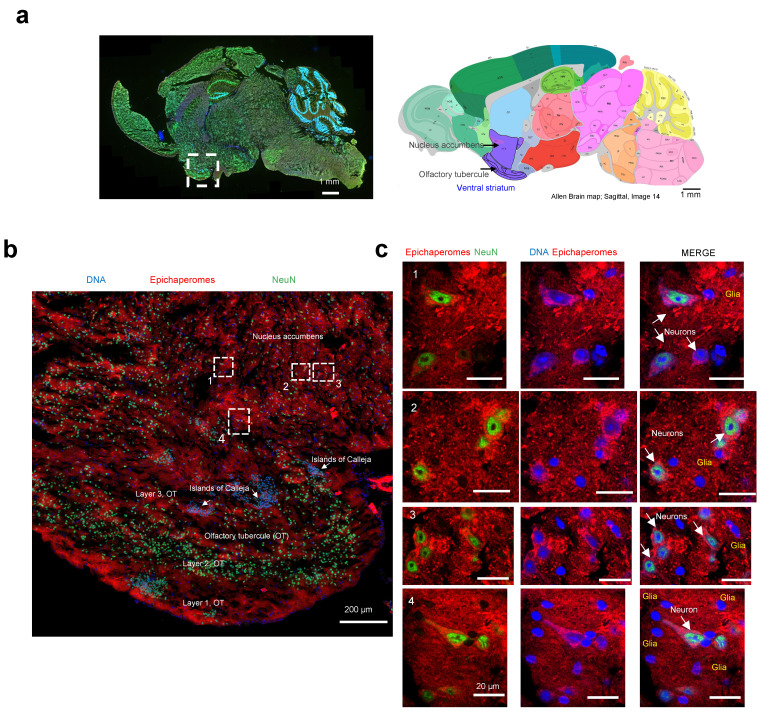
Evaluation of single-cell-level epichaperome illumination by PU-TCO in post-mortem murine brains. (**a**) **Left** side: Representative brain slice of male M83 homozygous mice (13 months of age) stained with Hoechst (blue) to detect individual brain cells and with an antibody against NeuN (Neuronal Nuclei, green) to discriminate neurons from glia. The approximate location of the ventral striatum is shown. **Right** side: Images for the anatomical location of brain areas closest to those used for epichaperome detection were obtained from the Allen Institute for Brain Science, Mouse, P56. Ventral striatum is highlighted in purple. Arrows point to the location of the nucleus accumbens and the olfactory tubercule. (**b**) Slices stained with PU-TCO (1 µM) and clicked to cy5, as shown in Figure 6, were re-imaged using a high-resolution microscope (Airyscan, Zeiss) to detect epichaperomes in individual cells. The micrograph shows the brain region encompassing the ventral striatum. The approximate location of its two subregions, the olfactory tubercle (OT), and the nucleus accumbens, as well as small clusters of neurons located within the ventral striatum of the brain, the Islands of Calleja, specifically found within the OT, are also shown. (**c**) Cell clusters, chosen from regions illustrated in panel (**b**), show the individual cells susceptible to epichaperome formation. The PU-TCO probe indicates that both neurons and glia are affected by epichaperomes (red). Blue represents Hoechst staining, while green represents NeuN.

## Data Availability

The original contributions presented in the study are included in the Appendix A, further inquiries can be directed to the corresponding author.

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
