# Peer review of "Synthesis and Characterization of Click Chemical Probes for Single-Cell Resolution Detection of Epichaperomes in Neurodegenerative Disorders"

_biomedicines, 2024, doi:10.3390/biomedicines12061252_

Round 1
Reviewer 1 Report
Comments and Suggestions for Authors
As per Sadik Bay et al, the manuscript reports the synthesis and characterization of clickable epichaperome probes, PU-TCO and PU-NTCO. Through comprehensive in vitro assays and cell-based investigations, authors established the specificity of these probes for epichaperomes. Furthermore, they demonstrated the efficacy of PU-TCO in detecting epichaperomes in brain tissue with cellular resolution, underscoring its potential as a valuable tool for dissecting single-cell responses in neurodegenerative diseases. However, the comments mentioned below need to be addressed.
· Introduction is too lengthy; authors may need to be more precise if possible.
· In the first paragraph of introduction, …..such as HSP90 and HSC70 [6],…. Mention the full form of these proteins here, but not in the 3rd paragraph.
· Authors mentioned “Leveraging a diverse range of commercially available fluorophores, such as far-red and near-infrared dyes [31], these probes offer superior tissue penetration and enhanced sensitivity, crucial for detecting epichaperomes within complex biological environments”. May be this statement sounds contradictory as authors didn’t perform any experiments with commercially available far-red and near-infrared dyes, and compare those results with clickable probes.
· 2.3.2 onwards, for room temperature it is referred to as rt without providing abbreviation, and in some cases (section 2.3.5), it referred as RT. May need to be consistent in all other sections wherever it is referring.
· Citation/explanation of Fig.1B is missing the results section.
· Fig. 3b, the western blot figures represent duplicates of one concentration of PU-TCO? Or each represents 1µM and 2µM of PU-TCO? Proper labeling should be provided.
· ….MDA-MB-468 and ASPC1 cells (Figure 5)… There are three subfigures fig.5a, 5b, and 5c. Which one authors referring to here? Need to specify it.
· What is Fig.5A? corresponding explanation is missing in results section. Instead of explaining more figure legends, it may be appropriate to mention them in the main paragraphs.
· ….wild-type (WT) mice at 13 months of age (Figure 6). Are you referring to 6a, or 6b? Need to be specific.
· … for epichaperomes both in vitro and in cells… What is the difference between in vitro studies and cell culture studies?
Author Response
As per Sadik Bay et al, the manuscript reports the synthesis and characterization of clickable epichaperome probes, PU-TCO and PU-NTCO. Through comprehensive in vitro assays and cell-based investigations, authors established the specificity of these probes for epichaperomes. Furthermore, they demonstrated the efficacy of PU-TCO in detecting epichaperomes in brain tissue with cellular resolution, underscoring its potential as a valuable tool for dissecting single-cell responses in neurodegenerative diseases.
Response: We sincerely appreciate the reviewer's thorough assessment of our study. It is gratifying to hear that our manuscript has been deemed comprehensive and robust in its ability to establish the specificity of the probes and their efficacy in detecting epichaperomes in brain tissue with cellular resolution. We thank the reviewer for valuable feedback, which undoubtedly has helped us improve the manuscript. Below, we address each comment provided.
Reviewer 1.1. Introduction is too lengthy; authors may need to be more precise if possible.
Response: Thank you for your feedback regarding the length of the introduction. We appreciate your perspective and would like to provide some insight into the rationale behind its structure. The introduction of our manuscript aims to provide readers with a comprehensive understanding of the concept of epichaperomes, which is a relatively new and evolving area of research. Given its novelty, it's essential to lay a solid foundation by elucidating the fundamental principles and significance of epichaperomes in the context of our study.
Furthermore, it's crucial to introduce readers to the existing methods used to study epichaperomes and why these methodologies may not be sufficient for addressing the specific research questions at hand. By discussing the limitations of current techniques, we aim to highlight the need for innovative approaches and methodologies, which our study seeks to address.
While we understand the importance of conciseness, we believe that providing this contextual background is essential for readers to grasp the significance of our research and its contribution to the field. However, at the reviewer’s recommendation, we have endeavored to streamline the introduction further without compromising clarity or depth.
Reviewer 1.2. In the first paragraph of introduction, …..such as HSP90 and HSC70 [6],…. Mention the full form of these proteins here, but not in the 3rd paragraph.
Response: fixed as suggested.
Reviewer 1.3. Authors mentioned “Leveraging a diverse range of commercially available fluorophores, such as far-red and near-infrared dyes [31], these probes offer superior tissue penetration and enhanced sensitivity, crucial for detecting epichaperomes within complex biological environments”. May be this statement sounds contradictory as authors didn’t perform any experiments with commercially available far-red and near-infrared dyes, and compare those results with clickable probes.
Response: It seems the reviewer may have misunderstood the statement regarding the use of diverse fluorophores in the context of clickable probes. The phrase "leveraging a diverse range of commercially available fluorophores" refers to the potential for incorporating various fluorescent tags into the clickable probes, not necessarily performing experiments with far-red and near-infrared dyes in the current study. We have reworded the text to improve clarity.
Reviewer 1.4. 2.3.2 onwards, for room temperature it is referred to as rt without providing abbreviation, and in some cases (section 2.3.5), it referred as RT. May need to be consistent in all other sections wherever it is referring.
Response: We fixed and spelled out the acronyms throughout the text
Reviewer 1.5. Citation/explanation of Fig.1B is missing the results section.
Response: Explanatory text was added as suggested (page 9).
Reviewer 1.6. Fig. 3b, the western blot figures represent duplicates of one concentration of PU-TCO? Or each represents 1µM and 2µM of PU-TCO? Proper labeling should be provided.
Response: The figure legend was revised for clarity.
Reviewer 1.7. MDA-MB-468 and ASPC1 cells (Figure 5)… There are three subfigures fig.5a, 5b, and 5c. Which one authors referring to here? Need to specify it.
Response: The text was revised for clarity (pages 12,13).
Reviewer 1.8. What is Fig.5A? corresponding explanation is missing in results section. Instead of explaining more figure legends, it may be appropriate to mention them in the main paragraphs.
Response: Text was revised to include the relevant explanation regarding Figure 5a (pages 12,13)
Reviewer 1.9. ….wild-type (WT) mice at 13 months of age (Figure 6). Are you referring to 6a, or 6b? Need to be specific.
Response: We revised the text for clarity (page 14).
Reviewer 1.10. … for epichaperomes both in vitro and in cells… What is the difference between in vitro studies and cell culture studies?
Response: Text was revised to include clarification (i.e., in vitro, in cell homogenates, and in cellulo, in live cells).
Reviewer 2 Report
Comments and Suggestions for Authors
This study introduces a new technology for detecting epichaperomes with high precision. Epichaperomes are disease-associated pathologic scaffolds composed of tightly bound chaperones, co-chaperones, and other factors. In this paper, the authors described the principles behind detecting epichaperomes using the PU-TCO epichaperome probe and the PU-NTCO control probe.
The manuscript provides very detailed descriptions, which are relevant to the field of molecular neuroscience and biochemistry and are presented in a well-structured manner.
The cited references are mostly recent and relevant, including highly impactful publications in the Nature Publishing Group, Neuron, etc.
The experimental design is appropriate for developing a clickable epichaperome probe and characterizing this probe with a relevant negative control.
The methods are given in detail, and the achieved results can be reproduced by other authors.
The paper is perfectly illustrated. The conclusions are consistent with the evidence presented in the Results section. The authors technique represents "a significant step forward in our understanding of epichaperome biology and its implications for neurodegenerative diseases" (Conclusion).
Author Response
This study introduces a new technology for detecting epichaperomes with high precision. Epichaperomes are disease-associated pathologic scaffolds composed of tightly bound chaperones, co-chaperones, and other factors. In this paper, the authors described the principles behind detecting epichaperomes using the PU-TCO epichaperome probe and the PU-NTCO control probe.
The manuscript provides very detailed descriptions, which are relevant to the field of molecular neuroscience and biochemistry and are presented in a well-structured manner.
The cited references are mostly recent and relevant, including highly impactful publications in the Nature Publishing Group, Neuron, etc.
The experimental design is appropriate for developing a clickable epichaperome probe and characterizing this probe with a relevant negative control.
The methods are given in detail, and the achieved results can be reproduced by other authors.
The paper is perfectly illustrated. The conclusions are consistent with the evidence presented in the Results section. The authors technique represents "a significant step forward in our understanding of epichaperome biology and its implications for neurodegenerative diseases" (Conclusion).
Response: Thank you to Reviewer 2 for the thorough and insightful assessment of our manuscript. We sincerely appreciate your recognition of the significance of our study in introducing a new technology for detecting epichaperomes with high precision. Your positive feedback on the detailed descriptions, relevance to the field of molecular neuroscience and biochemistry, and the well-structured presentation of our findings is invaluable to us.
We are also grateful for your acknowledgment of the appropriateness of our experimental design, the detailed methods provided, and the quality of the illustrations in the manuscript. Your recognition of the relevance and impact of the cited references further strengthens our confidence in the validity and significance of our work.